# Microglia and Astrocytes in Amyotrophic Lateral Sclerosis: Disease-Associated States, Pathological Roles, and Therapeutic Potential

**DOI:** 10.3390/biology12101307

**Published:** 2023-10-03

**Authors:** Justin You, Mohieldin M. M. Youssef, Jhune Rizsan Santos, Jooyun Lee, Jeehye Park

**Affiliations:** 1Genetics and Genome Biology Program, The Hospital for Sick Children, Toronto, ON M5G 0A4, Canada; justin.you@mail.utoronto.ca (J.Y.); mohieldin.youssef@sickkids.ca (M.M.M.Y.); jhunerizsan.santos@mail.utoronto.ca (J.R.S.); jooyun.lee@mail.utoronto.ca (J.L.); 2Department of Molecular Genetics, University of Toronto, Toronto, ON M5S 1A1, Canada

**Keywords:** amyotrophic lateral sclerosis (ALS), microglia, astrocyte, neurodegeneration, SOD1, TDP-43, C9ORF72, clinical trials

## Abstract

**Simple Summary:**

Amyotrophic lateral sclerosis (ALS) is a disease characterized by the death of motor neurons that results in progressive muscle weakness and paralysis. Therefore, much of ALS research has heavily centered on addressing the role of motor neurons in explaining the ALS disease process. However, accumulating evidence has suggested that non-neuronal cells, such as microglia and astrocytes, are actively involved in the disease process. This is currently an active field of research, and we sought to review the landmark discoveries in animal and human cell models regarding the important functions of microglia and astrocytes in ALS, how researchers define their various disease characteristics, and how these cells can be targeted for therapeutics.

**Abstract:**

Microglial and astrocytic reactivity is a prominent feature of amyotrophic lateral sclerosis (ALS). Microglia and astrocytes have been increasingly appreciated to play pivotal roles in disease pathogenesis. These cells can adopt distinct states characterized by a specific molecular profile or function depending on the different contexts of development, health, aging, and disease. Accumulating evidence from ALS rodent and cell models has demonstrated neuroprotective and neurotoxic functions from microglia and astrocytes. In this review, we focused on the recent advancements of knowledge in microglial and astrocytic states and nomenclature, the landmark discoveries demonstrating a clear contribution of microglia and astrocytes to ALS pathogenesis, and novel therapeutic candidates leveraging these cells that are currently undergoing clinical trials.

## 1. Introduction

Amyotrophic lateral sclerosis (ALS) is a progressive neurodegenerative disease characterized by the degeneration of motor neurons, resulting in muscle weakness, paralysis, and death [1]. Motor neurons are the largest neurons in the spinal cord, whose long peripheral nerves innervate muscle tissue. Since the discovery of the loss of motor neurons in ALS patients by Jean-Martin Charcot about 150 years ago, many research groups have been keen to answer why these motor neurons undergo death, and how we could restore the function and health of these neurons. Although the cause of ALS is unknown in most patients, 5–10% of ALS cases are due to inherited genetic mutations [2]. So far, about 30 genes are known to cause ALS [3,4]. Studies, particularly from genetic animal models, have provided an important insight into our understanding of the molecular mechanism underlying neurodegeneration, implicating that oxidative stress, mitochondrial dysfunction, axonal transport dysfunction, impaired protein homeostasis, and aberrant RNA metabolism may contribute to the degenerative process. From these genetic animal studies, we have also learned that other non-neuronal cells, such as microglia and astrocytes, play a key role in the neurodegenerative process starting at the early disease stage [5,6,7,8,9]. It is now evident that microglia and astrocytes exhibit neuroprotective and neurotoxic roles in genetic models not only in ALS but also in other neurodegenerative diseases [10,11,12,13,14,15]. The recent advent of single-cell multi-omics has revealed that microglia and astrocytes are highly plastic and can adopt distinct states in neurodegenerative diseases [16,17]. Currently, studies exploring the functional roles of each different state are actively being conducted. Here, we review the physiological functions of microglia and astrocytes in maintaining central nervous system (CNS) homeostasis and the fundamental discoveries demonstrating the beneficial and detrimental roles, mainly from the studies using ALS rodent and cell models. We also consider the recent recommendations on defining distinct microglial and astrocytic states, particularly in neurodegenerative diseases. Lastly, we outline the drugs that target microglia or astrocytes, which are currently being tested in clinical trials for ALS.

## 2. Microglia and Astrocytes: Their Physiological Roles and Disease-Associated States

### 2.1. Microglia

Microglia are the primary immune cells of the CNS parenchyma and comprise approximately 10% of cells in the CNS under normal physiological conditions [18,19]. Microglia were previously thought to originate from circulating monocytes due to their shared expression of immune markers with macrophages and monocytes [20]. However, evidence from fate-mapping studies suggests that microglia originate from c-Kit^+^ erythromyeloid progenitors in the yolk sac that migrate through the bloodstream to the developing CNS, starting at around embryonic day 8.5 in mice and continue until the blood–brain barrier is formed [21,22].

As the primary CNS immune cells, microglia are the first responders to sense and eliminate microbes invading the CNS parenchyma [23]. However, there is accumulating evidence that microglia have crucial physiological roles in CNS development and health (Figure 1). For instance, microglia regulate synapse remodelling and maturation by performing activity-dependent synaptic pruning in the developing and postnatal brain [24,25,26]. Microglia also play important roles in embryonic and adult neurogenesis by engulfing neuronal precursor cells (NPCs) to regulate the neuronal pool size [27,28]. Similarly, microglia can promote the development of oligodendrocytes by regulating the oligodendrocyte progenitor cell (OPC) pool [29]. Furthermore, microglia are directly involved in myelin turnover in the adult brain by phagocytosing myelin [30,31]. Recently, it has also been shown that microglia can sense extracellular ATP released by neurons and suppress further activation, thereby protecting the CNS against excess neuronal activity [32].

Microglia are relatively long-lived and able to self-renew, supporting the maintenance of a stable microglial network during homeostasis [20,33,34,35]. Microglia are morphologically characterized by their small cell body (around 7–10 μm) and highly ramified processes under homeostatic conditions [18]. These long processes are responsible for constantly surveying the CNS parenchyma for various cues, including pathogens, tissue injury, apoptotic cells, extracellular protein aggregates, myelin debris, and lipid degradation by-products. Microglia generally react to these extracellular triggers by altering their morphology, resulting in an enlarged cell body with shortened and fewer processes [18,19]. However, the morphological changes exhibited by microglia do not necessarily indicate any specific state or function [19].

Recently, a large international team of microglial experts has provided consensus recommendations regarding the nomenclature of microglial states [16]. It has been advocated against referring to microglia broadly into dichotomic categories that have been previously widely used [16]. For example, microglia under homeostatic conditions have often been referred to as resting microglia, whereas microglia that change their morphology in reaction to environmental cues have been termed activated microglia. Two-photon imaging of microglia revealed that their processes are highly mobile, extending and retracting at an average velocity of approximately 1.5 μm/min [36,37]. Microglia are now accepted as one of the most dynamic cells within the CNS and are never truly ‘resting’ [38]. Other terms widely used to dichotomize microglia are M1 (classically activated, pro-inflammatory, and neurotoxic) and M2 (alternatively activated, anti-inflammatory, and neuroprotective) states. However, it is evident from the expression profiling of microglia from several mouse models of disease that these cells do not polarize towards an M1 or M2 state and often express both M1 and M2 markers [16,39].

With the widespread use of single-cell technologies of microglia in various contexts, we now know that microglia can coexist as different multidimensional states with unique functions that change throughout development, health, injury, and disease [16]. Consequently, there have been various acronyms proposed to label these different microglial states. Notable examples include disease-associated microglia (DAM) [9], microglial neurodegenerative phenotype (MGnD) [40], activated response microglia (ARM) [41], interferon-responsive microglia (IRM) [41], lipid droplet-accumulating microglia (LDAM) [42], human Alzheimer’s disease microglia (HAM) [43], microglia inflamed in multiple sclerosis (MIMS) [44], RIPK1-regulated inflammatory microglia (RRIM) [45], glioma-associated microglia (GAM) [46], white matter-associated microglia (WAM) [31], proliferative region-associated microglia (PAM) [47], and axon tract-associated microglia (ATM) [48]. Notably, the functional implications of these microglial states and their relationship with each other represent an active field of research.

The microglial state that has received particular attention is DAM, as it was one of the first discovered microglial disease states. DAM signatures have been detected in several neurodegenerative diseases, including Alzheimer’s disease (AD), multiple sclerosis (MS), and ALS [9,40,49,50,51]. *Clec7a*, *CD9*, *Itgax* (*CD11c*), and *CD63* have been identified as potential DAM markers [9]. DAM show higher expression of genes implicated in phagocytic, lysosomal, and lipid metabolism pathways [52]. DAM also exhibit an increased expression of several genes linked to AD, such as *Apoe*, *Trem2*, *Lpl*, *Tyrobp*, and *Ctsd* [52].

The discovery that specific genes are required for microglia to transition into DAM revealed that DAM generally play a protective role in several neurodegenerative disease models. For instance, TREM2 is uniquely expressed in microglia and is essential in regulating the transition of microglia into a DAM state [9,40]. Indeed, several recent studies have demonstrated that the absence of TREM2 results in the inability of microglia to transition to a DAM-like state (including DAM, MGnD, and WAM) in neurodegenerative disease models or aged mice [9,40,53]. Deletion of TREM2 further increased the amyloid-β (Aβ) burden and accelerated the loss of cortical neurons in AD mice [54]. Knockout of TREM2 increased pathological inclusions of TDP-43 and worsened motor dysfunction and neurodegeneration in mice injected with ALS-linked TDP-43 [53]. In a mouse model of chronic demyelination, TREM2 deficiency caused the failure of microglia to clear phagocytosed myelin, resulting in pathogenic lipid accumulation [55]. However, in the case of tauopathy, the consequences of TREM2 KO are inconsistent, showing either a detrimental or protective effect depending on the specific tauopathy model used [56,57,58,59,60]. A recent study suggested that TREM2 may protect against tau pathology accumulation and subsequent neurodegeneration in models exhibiting both amyloid and tau pathology [56]. In addition to TREM2, SYK has been demonstrated to be essential for the microglial transition into the DAM state [61,62]. Microglial-specific deletion of *Syk* exacerbated Aβ burden, neuropathology, and cognitive defects in AD mice [61,62]. This neuroprotective role of microglial SYK was also demonstrated in mouse models of MS [62]. Together, these findings demonstrate that microglia play an important role in the pathogenesis of neurodegenerative diseases. Furthermore, DAM or a DAM-like state may be largely protective in neurodegenerative diseases by promoting the phagocytosis of pathological protein aggregates and myelin debris.

### 2.2. Astrocytes

Astrocytes comprise the largest population of glial cells, constituting approximately 30% of the cells in the mammalian CNS [63]. Astrocytes perform various essential functions to maintain proper development and health (Figure 2). Astrocytes play an instrumental role in neural synapse regulation by communicating with the pre-synaptic and post-synaptic membranes, generating an interconnected functional unit at the synapse, often called the tripartite synapse model [64]. Astrocytes respond to neurotransmitters and release gliotransmitters that regulate neuronal synaptic activity [65]. The released gliotransmitters, such as glutamate, ATP, GABA, or D-serine, induce distinct forms of synaptic regulation orchestrated via astrocytes [66], thus regulating neuronal network activity and information processing [67,68,69,70]. Astrocytes also perform neuroprotective functions by clearing excess neurotransmitters [71]. Indeed, astrocytes express transporters that facilitate neurotransmitter clearance at the synaptic cleft, such as the glial-specific glutamate transporters EAAT1 and EAAT2 [72] and the GABA transporters GAT-1 and GAT-3 [73,74]. In addition, they play a crucial role in maintaining the integrity of the blood–brain barrier and establishing a link between the bloodstream and the neurons [75]. Neuronal energy metabolism is regulated via astrocytes in a process termed the astrocyte–neuron lactate shuttle [76]. Astrocytes increase their rate of glucose utilization in response to neuronal activity and release lactate as an energy substrate for neurons [76].

Similar to microglia, astrocytes undergo robust morphological and functional changes in response to external stimuli, such as injury, infection, or disease [17,77]. Reactive astrocytes can exhibit protective roles through mediating proliferation, scar formation, blood-brain barrier repair, and immune cell recruitment [77,78,79,80]. This subtype of astrocytes has been traditionally referred to as A2 astrocytes. Conversely, A1 astrocytes are a subtype of reactive astrocytes induced by microglial factors, such as IL-1α, TNF-α, and C1q, and contribute to decreasing neuronal survival, outgrowth, and synaptogenesis [81]. Hence, the A1 astrocytes have been traditionally called neurotoxic due to their impact on neuronal death. Aside from the functional transformation, A1 astrocytes demonstrate a distinct molecular signature, with a hallmark of increased expression of the complement protein C3. Similar to microglial nomenclature recommendations [16], a large group of astrocyte researchers has provided a consensus on the nomenclature and definitions of reactive astrocytes [17]. Indeed, the strict dichotomous classifications of A1 (neurotoxic) and A2 (neuroprotective) astrocytes are now advocated against, as these may not represent true astrocytic diversity. Instead, astrocytic reactive states should be described based on their molecular expression patterns, functional changes, and impact on disease in relevant models [17].

Reactive astrocytes have been demonstrated to be implicated in neurodegenerative disorders, such as AD, Parkinson’s disease (PD), Huntington’s disease (HD), MS, and ALS (published by the authors of [82,83,84,85], which were reviewed by the authors of [86]). The neurotoxic effect of astrocytes in neurodegenerative diseases is mediated through exacerbated inflammatory and decreased neuronal support pathways [87]. Complement and immune pathways are aberrantly increased in reactive astrocytes, where the increased proportion of C3^+^ astrocytes was observed in patients with AD, ALS, PD, and HD [81]. The functions of astrocytes in regulating osmolarity and fluid clearance are altered in AD and ALS through decreased AQP4 levels [88,89]. Additionally, decreased expression of EAAT2 leading to dysfunctional astrocytes for glutamate recycling has been observed in ALS, AD, and PD, which might impart neuronal toxicity through increased excitability [72,90,91]. The transcriptomic profile for astrocytes in AD and ALS was altered, which may reflect their functional abnormalities [92,93,94,95]. 

## 3. Microglial and Astrocytic Reactivity in Human ALS

Microglial reactivity is a hallmark of human ALS. In post-mortem studies of ALS patients, an increase in IBA1^+^ and CD68^+^ microglia were observed near motor neurons in the corticospinal tract [96]. Microglial pathology was correlated with upper motor neuron deficits assessed using spasticity and reflex tests [96]. Similarly, transcriptomic analysis of the motor cortex revealed an overrepresentation of microglia in ALS compared to healthy controls, in which a subpopulation showed gene signatures overlapping with DAM [97]. Additionally, positron emission tomography (PET) revealed increased microglial reactivity in the motor and extra-motor cerebral regions, cerebellum, thalamus, and medulla oblongata of ALS patients [98,99]. These results indicate that microglial reactivity is specifically present in regions affected by ALS, further supporting the notion that microglia may contribute to ALS pathology.

Similar to microglial reactivity, reactive astrogliosis is a key feature in ALS [100,101,102]. Furthermore, RNA sequencing of 380 post-mortem spinal cord samples showed increased gene expression associated with reactive astrocytes [103]. To visualize astrogliosis in vivo through PET imaging, tracers targeting monoamine oxidase-B (MAO-B), a protein that is highly expressed in the outer mitochondrial membrane in reactive astrocytes [17,104], have been used in a variety of diseases, including ALS [105]. PET imaging of ALS patients showed increased uptake of deuterium-substituted [^11^C](L)-deprenyl-D2, a tracer that selectively binds to MAO-B [105], demonstrating astrogliosis in the pons and white matter of ALS patients [106]. Together, these results demonstrate that microglia and astrocytes react to pathology in human ALS and may contribute to the disease pathogenesis.

## 4. Role of Microglia and Astrocytes in ALS Rodent Models

### 4.1. Glial Studies from SOD1 Rodent Models

Superoxide dismutase 1 (SOD1) was the first gene discovered to be linked to ALS in 1993 [107], and has been found to comprise around 20% of familial cases [108]. Furthermore, several different SOD1 mutant mice have been shown to recapitulate many key features of human ALS [109,110,111,112] and currently represent the most widely used model for ALS. There has been early evidence to suggest an important non-neuronal component for ALS in mutant SOD1 mouse models. Early reports showed that the neuron-specific expression of SOD1 G37R, G85R, or G93A was insufficient to induce ALS phenotypes [113,114]. However, some conflicting evidence has shown that the neuron-specific expression of mutant SOD1 is sufficient to cause ALS [115,116]. Although, it is important to emphasize that the effects of mutant SOD1 in these studies on motor function and survival are modest compared to mouse models that ubiquitously express mutant SOD1. Furthermore, wild-type non-neuronal cells were found to delay degeneration and extend the survival of mutant SOD1-expressing motor neurons [117]. Together, these results suggest that non-neuronal cells may be the predominant contributor to ALS progression in SOD1 mutant mice.

Additional studies have investigated the role of microglia in ALS pathogenesis in SOD1 mutant mice. It should be noted that many of these studies have used outdated terms when referring to different microglial states [16], and we will refrain from using them in this text. Interestingly, SOD1 G93A mice were used in the initial studies that demonstrated that microgliosis results from an expansion of resident microglia as opposed to circulatory progenitors [35]. In comparison to the SOD1 G93A-expressing microglia, wild-type donor-derived microglia were shown to slow down neurodegeneration and increase survival in SOD1 G93A mice that lack macrophages, neutrophils, T cells, B cells, and microglia [118]. Primary microglia from SOD1 G93A mice exhibited an increased production of reactive oxygen species (ROS) and nitric oxide (NO), as well as a decreased production of neurotrophic factors, such as IGF1, which coincided with a decreased survival of co-cultured primary neurons [118,119]. Furthermore, the addition of an inducible NO synthase (iNOS) inhibitor rescued motor neurons from microglia-mediated toxicity [119]. The selective inhibition of NF-κB in microglia rescued the degeneration of motor neurons in vitro and extended survival in SOD1 G93A mice by blunting pro-inflammatory responses [120]. Treating SOD1 G93A mice with a selective CSF1R inhibitor, which reduces microglial cell proliferation, demonstrated decreased motor neuron cell death and extended survival [121]. However, targeting microglial proliferation yields inconsistent results, which may be attributable to the method used. For instance, another study used transgenic mice, which allowed for the elimination of proliferating microglia following the administration of ganciclovir [122]. This resulted in a drastic reduction in the number of microglia in the lumbar spinal cord in SOD1 G93A mice, but did not affect motor neuron degeneration [122].

Studies have shown that microglia display neuroprotective or neurotoxic roles depending on the disease stage. Isolated microglia from SOD1 G93A mice at disease onset enhanced motor neuron survival when co-cultured with motor neurons, whereas microglia from end-stage mice were toxic to motor neurons [123]. These findings were also supported by another study that revealed a drastic upregulation of the anti-inflammatory cytokine IL-10 from primary microglia isolated from SOD1 G37R mice at the pre-symptomatic stage [124]. Furthermore, overexpression of IL-10 in microglia delayed disease onset and increased survival in SOD1 G93A mice [124]. In addition, reducing the expression of mutant SOD1 in microglia delayed disease progression in the late disease stage of SOD1 G37R mice, but not in the early disease stage [125]. 

Studies have shown that reactive astrocytes are involved in ALS pathogenesis in SOD1 mutant mice, evidenced by a progressive increase in astrogliosis in SOD1 mutant mice as the disease progresses [126]. Previous studies have demonstrated that astrocytes play a toxic role in disease pathogenesis. Deleting mutant SOD1, specifically in astrocytes, delayed disease progression in transgenic SOD1 mutant mice [126,127]. Similarly, transplantation of wild-type astrocyte precursors in the cervical spinal cord of SOD1 G93A mice extended their survival, slowed the decline in motor performance, and partially prevented motor neuron loss in these mice [128]. Moreover, astrocyte-selective RNA interference against mutant SOD1 G93A significantly improved the neuromuscular function of SOD1 G93A mice and rescued a small population of fast-fatigable motor neurons [129]. Conversely, transplantation of SOD1 G93A astrocyte precursors in wild-type mice reduced neuromuscular functions and induced motor neuron death [130]. In contrast to these findings, astrocyte-restricted expression of SOD1 G86R on mice only resulted in changes to astroglial morphology, but did not result in motor deficits or motor neuron degeneration [131]. Interestingly, this in vivo study did not recapitulate previous reports, suggesting that additional studies are required for further exploration of astrocytes in ALS.

The toxicity of astrocytes to motor neurons has also been established from in vitro experiments. Co-cultures of astrocytes expressing mutant SOD1 with motor neurons impaired the survival of motor neurons expressing either wild-type or mutant SOD1 [132,133]. Interestingly, the toxic effects of astrocytes appear to be specific to motor neurons, as a similar co-culture of mutant astrocytes with other neuronal cell types, including dorsal root ganglion neurons and interneurons, did not induce significant neuronal death [134]. Another study involving a human embryonic stem cell-derived neuronal–glia co-culture system supported that glial toxicity is specific to motor neurons [135]. Furthermore, this mutant astrocyte toxic effect was suggested to be mediated through the release of soluble factors, as the conditioned media (CM) from mutant astrocytes displayed a similar toxicity to motor neurons [132,134]. Intriguingly, these toxic mediators seem to be specific to astrocytes, as CM from myocytes, microglia, cortical neurons, or skin fibroblasts expressing similar levels of mutant SOD1 did not display comparable toxicity compared to the mutant astrocyte CM [134]. A recent study demonstrated that the excessive release of inorganic polyphosphates (poly-P), a gliotransmitter from astrocytes, is a toxic factor that leads to neuronal hyperexcitability and subsequent neuronal death [136,137]. Notably, this gliotransmitter has also been reported to display pro-inflammatory properties outside the CNS [138]. Another potential identity of a neurotoxic soluble factor is ROS, as an increase in ROS production occurs in SOD1 G93A astrocytes [139]. Furthermore, treating primary spinal cord cultures with CM from SOD1 G93A astrocytes induced oxidative stress and cell death in motor neurons [140]. Another neurotoxic factor may be TGF-β1, which is upregulated in astrocytes in mice models and human ALS [141]. The astrocyte-specific overproduction of TGF-β1 in SOD1 G93A mice was found to exacerbate disease progression [141]. There is also some evidence that ALS astrocytes may decrease the secretion of neurotrophic factors, which may also contribute to increased neurotoxicity. Primary astrocytes from SOD1 G93A mice, but not wild-type mice, showed decreased production of BDNF and GDNF in response to HMGB1, a nuclear protein released by cells undergoing damage or stress [142]. Another study showed that treating wild-type primary astrocytes with the CSF of sporadic ALS patients caused a decreased production of the neurotrophic factors VEGF and GDNF [143].

Although these studies have elaborated on the impact of astrocytes on motor neuron survival in SOD1-ALS, it is also likely that motor neurons can influence astrocytes in ALS. A recent study demonstrated that astrocytes can uptake the motor neuron-enriched microRNA 218 (miR-218) [144], which was previously identified to be released extracellularly from motor neurons in response to injury/death [145]. Uptake of miR-218 in mouse primary astrocytes resulted in the downregulation of the astrocytic glutamate re-uptake transporter EAAT2 [144], a key player in regulating excitatory glutamatergic neurotransmission. EAAT2 expression is markedly reduced in ALS post-mortem brain tissues, which could underlie neuronal excitotoxicity [146]. Indeed, overexpressing EAAT2 in SOD1 G93A mice protected neurons from L-glutamate-induced cytotoxicity and delayed muscle strength decline and motor neuron loss, but did not affect paralysis, body weight, or survival [147]. The downregulation of miR-218, through intracerebroventricular injections of miR-218 antisense oligonucleotides, in symptomatic SOD1 G93A mice restored EAAT2 expression and normalized GFAP and connexin 43 expression [144]. These findings may suggest the presence of a neurotoxic feedback loop—ALS astrocytes contribute to motor neuron death through the extracellular release of neurotoxic factors, and in turn, dying motor neurons release factors that influence astrocytes to further contribute to MN death (e.g., via excitotoxicity). 

The crosstalk between microglia and astrocytes has been recently found to contribute to neurodegeneration. The secretion of three microglial factors—IL-1α, TNF-α, and C1q—is sufficient to induce C3^+^ reactive astrocytes, resulting in the loss of several physiological functions, including impaired synapse formation and phagocytic capacity [81]. Furthermore, these reactive astrocytes were found to be highly neurotoxic and induce cell death through very-long-chain fatty acid acyl chains and long-chain saturated free fatty acids that are contained within reactive astrocyte membranes and APOE- and APOJ-containing lipoparticles [81,148]. Intriguingly, knocking out *IL-1α*, *TNFα*, and *C1q* dramatically lowered the levels of the reactive astrocyte marker *C3* in SOD1 G93A mice [149]. This resulted in a remarkable extension in survival by over 50%, and also led to a rescue in motor deficits and a significant delay in lower motor neuron degeneration [149].

There is also important crosstalk between the microglia and astrocytes with peripheral immune cells, including T cells and natural killer (NK) cells. The astrocyte-specific overproduction of TGF-β1 in SOD1 G93A mice accelerated disease progression and reduced T cell infiltration and the levels of T cell-related cytokines and neurotrophic factors [141]. Inversely, T cells have also been found to influence the number of microglia and astrocytes. SOD1 G93A mice bred with mice lacking functional T cells resulted in decreased survival and decreased the number of microglia and astrocytes in the lumbar spinal cord [150]. Another study showed that depleting NK cells in SOD1 G93A mice delayed motor deficits, increased survival, and modulated the inflammatory response of microglia and infiltration of regulatory T cells [151]. 

### 4.2. Glial Studies from TARDBP Rodent Models

Mutations in TAR DNA-binding protein (*TARDBP*), which encodes for TDP-43, were discovered to cause familial ALS in 2008 [152]. Interestingly, despite mutations in *TARDBP* being implicated in approximately 5% of familial ALS cases and approximately 1% of sporadic cases, mislocalization of TDP-43 in cytoplasmic aggregates has been observed in more than 95% of all ALS cases [153]. This suggests two possible mechanisms that may contribute to ALS pathogenesis: a toxic loss-of-function mechanism through the nuclear depletion of TDP-43 and a toxic gain-of-function mechanism through cytoplasmic aggregation. Cytoplasmic TDP-43 aggregates have been found in both neurons and glial cells, suggesting that TDP-43 may exert toxicity in both cell types [154,155,156].

Several studies have suggested that microglia play a beneficial role in TDP-43-mediated ALS pathogenesis. A mouse model in which human *TARDBP* (hTDP-43) expression within neurons is controlled in the presence of doxycycline has been utilized to investigate microglial responses to TDP-43-induced toxicity in neurons [157]. Upon suppression of hTDP-43 expression in neurons, microglial proliferation and morphological changes were induced and associated with increased TDP-43 clearance in neurons. Moreover, microglial depletion using a CSF1R inhibitor resulted in failure to regain motor function, suggesting a neuroprotective role of microglia in TDP-43 mice. Additionally, a recent study demonstrated the role of DAM in ALS pathogenesis in a viral-mediated and transgenic mouse model that overexpresses hTDP-43 [53]. TREM2 KO resulted in the ablation of DAM signatures of microglia, impaired microglial clearance of pathological TDP-43 inclusions, and worsened motor dysfunction and neurodegeneration in mice that express hTDP-43. Together, these findings suggest that microglia play a protective role in TDP-43-linked ALS.

Previous studies have suggested that the decreased or increased expression of TDP-43 exerts toxicity in astrocytes, leading to ALS-like phenotypes. Mice with an astrocyte-specific deletion of TDP-43 show mild motor deficits, but no significant pathology in motor neurons and neuromuscular junctions [158]. Notably, this loss of TDP-43 in astrocytes also increased GFAP immunoreactivity, and altered the astrocyte transcriptome to resemble that of pro-inflammatory astrocytes [158]. Similarly, knockdown of TDP-43 in primary rat astrocytes also resulted in transcriptomic changes reminiscent of reactive astrocytes [159]. Furthermore, astrocyte-specific expression of ALS-linked TDP-43 M337V upregulated GFAP expression and resulted in progressive paralysis and spinal motor neuron loss [160]. Interestingly, astrocyte-specific TDP-43 M337V expression led to a progressive depletion of the astrocytic glutamate transporters EAAT1 and EAAT2, and induced the expression of lipocalin-2 (lcn2) [160], a neurotoxic factor released by reactive astrocytes that is increased in the post-mortem ALS motor cortex and spinal cord [161]. This upregulation of lcn2 was also observed in reactive astrocytes from transgenic hTDP-43 rats [162]. Synthetic lcn2 exerts a dose-dependent cytotoxicity to primary neurons [162]. Additionally, overexpression of TDP-43 in mouse primary astrocytes elevated the secretion of the pro-inflammatory cytokines IL-1β, IL-6, and TNF-α [163]. Furthermore, astrocytic overexpression of TDP-43 induced neurotoxicity and mitochondrial dysfunction in differentiated NSC-34 cells, a motor neuron-like cell line, and in mouse primary cortical neurons. This toxicity was attenuated in the presence of neutralizing antibodies targeting IL-1β, IL-6, and TNF-α.

### 4.3. Glial Studies from C9ORF72 Rodent Models

The hexanucleotide repeat expansion of GGGGCC in *C9ORF72* was first reported to cause ALS in 2011 [164,165], and is currently known as the most common cause of familial ALS, found in around 25% of cases [108]. Three broad mechanisms have been proposed to explain how repeat expansions in *C9ORF72* could cause ALS. The first mechanism is a loss of C9ORF72 function, as reduced *C9ORF72* expression is displayed in ALS patients that carry hexanucleotide repeat expansions [166]. The second mechanism is a gain-of-function toxicity due to the presence of nuclear RNA foci observed in cells from *C9ORF72* expansion carriers [165], which have been proposed to cause alterations in RNA metabolism and splicing defects due to the sequestration of RNA-binding proteins [167]. The third mechanism is a gain-of-function toxicity due to the production of dipeptide repeat (DPR) proteins from repeat-associated non-AUG (RAN) translation found in the CNS of *C9ORF72* expansion carriers [168]. Research on *C9orf72* rodent models has provided evidence to support these mechanisms of *C9ORF72*-mediated ALS pathogenesis, and that these mechanisms may also affect non-neuronal cells, such as microglia and astrocytes.

Although all cell types express *C9orf72* in the brain, microglia exhibit the highest levels of *C9orf72* expression [169]. Interestingly, *C9orf72* KO mice do not display ALS-like features, such as neurodegeneration, motor function defects, or reduced survival [169,170,171]. Despite the lack of ALS phenotypes in these mice, the reduced expression of *C9orf72* caused alterations in microglia, as they exhibited lysosomal accumulation and increased levels of pro-inflammatory cytokines, such as IL-1β and IL-6 [169]. Additionally, a recent study showed that the microglial-specific deletion of *C9orf72* promoted synapse loss, neuronal deficits, and worsened memory, while paradoxically improving plaque clearance in AD mice [172]. Together, these findings implicate that C9ORF72 plays an important role in microglia, and that its loss in microglia may contribute to neuronal dysfunction.

Many groups have attempted to model *C9orf72*-mediated ALS using bacterial artificial chromosome (BAC) transgenic mouse models that carry the intronic repeat expansion mutation in *C9orf72* [170,173,174,175]. While these mutant mice display the histopathological features seen in ALS, only the model described by Liu and colleagues shows the neurodegenerative features and motor deficits seen in human *C9ORF72*-ALS [173,176]. The phenotypic variability observed in these transgenic mouse models may reflect the incomplete penetrance of the *C9ORF72*-ALS mutation in humans [177]. The model by Liu and colleagues, however, remains controversial, as it has been reported to have variable survival and motor phenotypes that were not recapitulated by at least one other group [178], which has been suggested to be due to methodological differences during the analysis, as well as possible genetic and environmental differences [179]. Nevertheless, the groups that did observe motor and survival phenotypes reported increased staining of IBA1^+^ microglia and GFAP^+^ astrocytes in the brains of the *C9orf72*-ALS mice [173,176,179]. Similarly, transgenic mice expressing DPRs (poly-GA, poly-GR, and poly-PR) display neurodegeneration and motor deficits accompanied by increased microglial and astrocyte proliferation [180,181,182,183,184]. Reducing the levels of these DPRs, as shown recently through poly-GA vaccination in poly-GA mice [185] and antibody therapy in C9-BAC mice [176], rescued motor deficits and reduced neuroinflammation. 

### 4.4. Insights from Human iPSC-Derived Microglia and Astrocytes That Model ALS

Although rodent models of ALS have provided an immense amount of knowledge regarding the roles of microglia and astrocytes in ALS pathogenesis, there are still several limitations. For instance, there are notable cross-species differences in both microglia and astrocytes [186,187,188]. Furthermore, most research investigating the pathological roles of microglia and astrocytes are performed on genetic models of ALS, despite most ALS cases being sporadic. Therefore, human-induced pluripotent stem cell (iPSC)-derived microglia and astrocytes from ALS patients have been an excellent complementary model.

The recent establishment of the protocol to generate microglia from human iPSCs [189] has provided an exciting novel approach to study the function of human microglia-like cells and their dysfunction in the context of neurodegenerative diseases. As ALS studies using human iPSCs are beginning to unfold, few studies have been published examining the effect of the *C9ORF72* mutant in iPSC microglia [190,191,192]. It appears that *C9ORF72* mutant microglia are insufficient in being able to cause overt toxicity to healthy motor neurons [191]. However, priming microglia with LPS caused a greater increase in pro-inflammatory responses in *C9ORF72* mutant microglia compared to that of the control [190,191]. Furthermore, LPS-primed *C9ORF72* mutant microglia decreased motor neuron survival, whereas unstimulated microglia showed no overt toxicity [191]. This neurotoxicity is partly mediated through the increase in the release of MMP9 in LPS-primed *C9ORF72* mutant microglia [191]. Another study showed an impairment in autophagy in *C9ORF72* mutant and KO iPSC-derived microglia compared to that of the control, which coincided with increased sensitivity of mutant *C9ORF72* iPSC-derived motor neurons to excitotoxic stimuli [192]. Additionally, monocyte-derived microglia have also been used to study the functions of microglia in sporadic ALS patients. Indeed, sporadic ALS microglia exhibited TDP-43 inclusions, abnormal phagocytic ability, and increased pro-inflammatory responses to LPS compared to that of the control [193,194]. 

Studies using iPSC-derived astrocytes have provided valuable insights into their role in ALS pathogenesis. Several studies have suggested that iPSC-derived astrocytes from ALS patients are more toxic to motor neurons compared to control astrocytes [195,196,197,198,199,200]. Mechanistically, there have been several proposed mechanisms for how ALS astrocytes mediate their neurotoxicity. For instance, ALS iPSC-derived astrocytes exhibit defective autophagy [195,201] that coincides with increased activation of pro-inflammatory pathways compared to control astrocytes [195]. ALS iPSC-derived astrocytes can also impair autophagic flux in HEK293T cells [202]. Furthermore, iPSC-derived astrocytes from *C9ORF72*-ALS patients show a reduction in the secretion of antioxidant proteins, which subsequently results in oxidative stress in both astrocytes and wild-type motor neurons [203]. *C9ORF72*-ALS-derived astrocytes show a downregulation of miR-494-3p, in which restoring the levels of miR-494-3p using an miRNA mimic increased motor neuron survival [197]. Additionally, TDP-43-ALS-derived astrocytes exhibit pathological hallmarks, such as cytoplasmic TDP-43 and increased poly-P levels [200]; interestingly, the neutralization of poly-P in this TDP-43-ALS-derived astrocyte CM prevented motor neuron death [200]. ALS-derived astrocytes also exhibit a loss of metabolic flexibility compared to that of their controls [198,204]. These studies have elucidated some mechanisms by which ALS-derived astrocytes induce neurotoxicity. Conversely, iPSC-derived astrocytes from healthy humans show characteristics of neuroprotective functions. iPSC-derived motor neurons treated with sporadic ALS post-mortem tissue extracts result in seeded aggregation of TDP-43 [205]. Interestingly, co-culturing these motor neurons with healthy iPSC-derived astrocytes reduced cytoplasmic TDP-43, TDP-43 aggregation, and cell toxicity [205].

## 5. Candidate Therapeutics for ALS Targeting Microglia or Astrocytes

There is currently no cure for ALS, with limited approved treatment options including riluzole, edaravone, Albrioza (AMX0035), and Qalsody (tofersen). However, these treatments only have modest effects on prognosis. For example, the recently approved drug Albrioza possesses several benefits, including slowing functional decline and extending survival by 6.5 months [206,207,208]. Although these drugs represent important increments for the future of ALS care, their mild clinical benefits emphasize the increased need for novel ALS therapeutics.

Although several other papers have reviewed the potential ALS drugs in clinical trials in recent years [209,210,211], we will review a non-exhaustive list of potential therapeutics that leverages microglia or astrocytes and have been tested in clinical trials of ALS in recent years. These therapeutics broadly target various aspects of microglial or astrocytic pathological states in ALS, which include small molecules/inhibitors that target pathways involved in immune cell survival, proliferation, inflammatory response, or cell-based transplantations. However, it is important to note that many of these therapeutics may not exclusively target microglia or astrocytes, but may also target various other immune cells involved in the pathogenesis of ALS.

### 5.1. Masitinib

Masitinib is an inhibitor of tyrosine kinase and has been demonstrated to be neuroprotective in SOD1 G93A rat models by primarily targeting microglia and mast cells [212,213,214]. Masitinib also slowed disease progression in a phase 2/3 clinical trial in ALS patients when used as an add-on therapy to riluzole [215]. Oral masitinib is undergoing a randomized, parallel-group, double-blind phase 3 clinical trial to test its efficacy and safety in combination with riluzole (NCT03127267). 

### 5.2. RIPK1 Inhibitors

RIPK1 is a serine/threonine kinase highly expressed by microglia and astrocytes within the CNS [45]. There is accumulating evidence that RIPK1 exerts a pro-inflammatory role in the CNS of ALS animal models [45,216,217]. Furthermore, RIPK1 inhibition reduced the number of microglia in a state referred to as RIPK1-regulated inflammatory microglia (RRIM) that produce pro-inflammatory cytokines in ALS mice, suggesting that RIPK1 inhibitors could be a potential ALS therapy [45]. Currently, a phase 2 clinical trial for evaluating the safety and effectiveness of the RIPK1 inhibitor SAR443820 is actively enrolling participants with ALS (NCT05237284).

### 5.3. CSF1R Inhibitors

The use of CSF1R inhibitors has shown conflicting results in preclinical ALS models. GW2580 is a potent inhibitor of CSF1R that reduces microglial proliferation and macrophage invasion into the peripheral nerves of SOD1 G93A mice [121]. Furthermore, it has been demonstrated to slow disease progression and extend survival in SOD1 G93A mice [121]. However, conflicting results were seen in an inducible hTDP-43 mouse model that can regain motor function after suppressing hTDP-43 expression [157]. Interestingly, mice that were treated with PLX3397, another CSF1R inhibitor, failed to regain full motor function [157]. Notably, PLX3397 was also found to exhibit inhibitory effects on c-Kit, whereas GW2580 was revealed to demonstrate a high selectivity for CSF1R [121]. Nonetheless, the effects of CSF1R inhibitors in ALS patients are still unknown. A phase 2 clinical trial is undergoing recruitment to determine the safety, tolerability, and microglial response to the CSF1R inhibitor BLZ-945 in ALS patients (NCT04066244).

### 5.4. MN-166 (Ibudilast)

MN-166 is a small molecule that is a potent inhibitor of macrophage migration inhibitory factor (MIF) and phosphodiesterases PDE3, PDE4, and PDE10 [218]. Additionally, this molecule exhibits neuroprotective effects by inhibiting the production of pro-inflammatory cytokines and nitric oxide species in microglia and astrocytes, as well as increasing the production of anti-inflammatory cytokines and neurotrophic factors [218]. Administering MN-166 to HEK293 cells that overexpress either SOD1 G93A or TDP-43 increased the clearance of SOD1 or TDP-43 aggregates [219]. A phase 2b clinical trial demonstrated no difference in glial reactivity in the brain from PET imaging or changes in serum neurofilament light levels [220]. However, MN-166 was deemed to have an acceptable safety profile [220]. There is ongoing recruitment for a phase 2b/3 clinical trial to assess for the potential efficacy of MN-166 in ALS (NCT04057898) [218].

### 5.5. Complement Cascade Inhibitors

Several drugs have been developed to target proteins in the complement cascade, a component of the innate immune system that is implicated in the pathogenesis of ALS [221]. However, the phase 3 trial (NCT04248465) to evaluate the safety and efficacy of the C5 inhibitor ravulizumab was discontinued due to a lack of efficacy in August 2021. Furthermore, zilucoplan is another C5 inhibitor for which the phase 2/3 trials were also discontinued due to a lack of efficacy (NCT04297683 and NCT04436497). Although these results suggest that inhibiting C5 may not be efficacious in ALS, other drugs are currently being tested that target different complement proteins. Pegcetacoplan is a C3 inhibitor for which a currently active phase 2 clinical trial is testing its safety and efficacy in ALS (NCT04579666). Additionally, there is ongoing recruitment for a phase 2 clinical trial to test the C1q inhibitor ANX005 in ALS (NCT04569435).

### 5.6. Astrocyte Transplantation

CNS10-NPC-GDNF is a cell-based therapeutic that refers to the neural progenitor cell line derived from a single human fetal cortical sample, and is transduced to express glial cell line-derived neurotrophic factor (GDNF) [222]. GDNF is a potent survival factor for motor neurons [223], but has a short plasma life and low CNS penetrance [224]. Therefore, transplantation of CNS10-NPC-GDNF allows for a stable production of GDNF, and has been demonstrated to show the preservation of motor neurons in SOD1 G93A rats [225,226]. Additionally, these transplanted cells exhibit long-term survival, and can differentiate into astrocytes and provide neuroprotection of motor neurons in the spinal cord of aged rats [227,228]. Recently, a phase 1/2a clinical trial met the safety endpoint following unilateral injections of CNS10-NPC-GDNF in the lumbar spinal cord of ALS patients, and maintained GDNF production up to 42 months after transplantation [222]. There is ongoing recruitment for another phase 1 clinical trial to determine the safety of CNS10-NPC-GDNF transplantation into the motor cortex of ALS patients (NCT05306457). Another cell-based therapeutic, AstroRX, represents human astrocytes that are derived from embryonic stem cells, and have been shown to recapitulate properties of healthy astrocytes, including the uptake of glutamate, promoting outgrowth of axons, and protecting motor neurons from oxidative stress [229]. Intrathecal injections of AstroRX in SOD1 G93A rodents significantly slowed disease onset and improved motor function [229]. Results from phase 1/2a clinical trials showed that a single intrathecal injection of AstroRX into ALS patients was well-tolerated [230]. Moreover, human glial-restricted progenitor cells, Q-cells, have been designed to differentiate into astrocytes after injection. Q-cells showed promising preclinical results in promoting the survival of dysmyelinated neurons in mice [231]. This cell-based approach is currently undergoing a phase 1/2 clinical trial for ALS treatment (NCT02478450).

## 6. Conclusions

Microglia and astrocytes have been increasingly appreciated to play essential roles throughout development, health, injury, and disease. Furthermore, their vast diversity and functions have been recently apparent through the widespread use of single-cell omics, revealing that these cells can assume distinct states depending on the context. The discovery of disease-associated states has enhanced our understanding of the biology of microglia and astrocytes. However, this came with an unintended consequence, more so in microglia, in which many different acronyms have been assigned to different cell states, which could lead to misleading interpretations. For instance, the true biological function of these cellular states may not be accurately reflected by the enriched biological pathways indicated by their gene signature. Furthermore, mRNA expression may not directly correlate with protein levels. Another limitation is that these acronyms may be interpreted as stable states of microglia, which may not necessarily be the case. Indeed, these cells are highly plastic and appear to change in response to varying contexts. Fortunately, large teams of experts have provided expert recommendations regarding the nomenclature of microglial and astrocytic states, highlighting the importance of providing nuance [16,17]. However, these nomenclatures are still not unanimously used in the field at large, as evident from many recent publications still using these overly simplistic binary terms, such as M1 vs. M2, and A1 vs. A2. It is still unclear as to the extent of microglial and astrocytic diversity in disease, which highlights the need for more research to determine the vast morphological, molecular, and functional heterogeneity of these cells.

Microglia and astrocytes undoubtedly have an important role in ALS pathogenesis, especially due to the accumulating evidence from ALS rodent and cell models. Indeed, there is strong evidence over many decades that microglial and astrocytic reactivity is a hallmark of human ALS. Evidence from ALS rodent and cell models has suggested that microglia can exhibit neuroprotective and neurotoxic effects (Figure 1), with their protective roles being more prominent in the early disease stages. In contrast, reactive astrocytes are generally neurotoxic in ALS, likely through a combination of secreted neurotoxic factors and loss of supportive functions (Figure 2). 

The development of future therapeutics should take a more holistic viewpoint of ALS pathology. Rather than exclusively focusing on the motor neuron-centric model of pathogenesis, which has been so often conducted in the past, there should be a strong emphasis placed on non-neuronal cells, such as microglia and astrocytes. Additionally, in vitro modelling using human iPSC-derived or primary mouse microglia and astrocytes should seek to combine multiple cell types of relevance, given the known crosstalk between multiple different cell types. Indeed, there are many promising therapeutics that target these cells, which are currently under investigation in clinical trials. However, given that various disease states exist in microglia and astrocytes, a potential approach for future therapeutics should target specific microglial or astrocytic disease states, for example, increasing the number of ‘good microglia’, possibly DAM, or targeting neurotoxic reactive astrocytes. More research is needed to determine optimal druggable targets to translate these findings into clinical models. Furthermore, considering the important immune functions of microglia and astrocytes, targeted therapeutics should also consider the potential risks associated with elevated susceptibility to infections.

Clearly, more research in this field is required to fully grasp the underlying biology of how these cells contribute to ALS. Several specific factors, such as sex [232], microbiota [233,234], diet [235,236], exposure to immune signals [237], and disease stage, could all affect the function of microglia and astrocytes and should be taken into consideration. Given that modulating microglial and astrocytic function can significantly impact ALS pathogenesis, deciphering the characteristics of various disease states and implementing new methods of selectively targeting these cell states will likely lead to promising therapeutic candidates.

## Figures and Tables

**Figure 1 biology-12-01307-f001:**
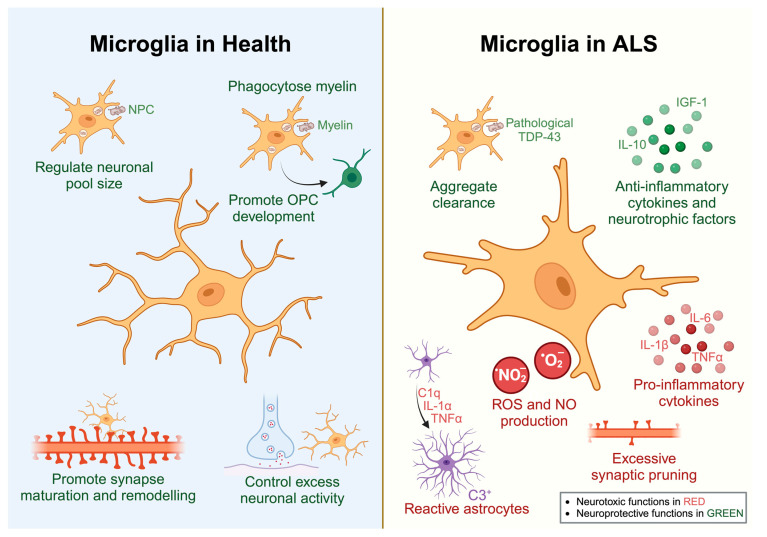
Physiological roles of microglia in health and ALS. Microglia play a myriad of functions to maintain neuronal health, including myelin phagocytosis, regulating neuronal pool size, synaptic pruning, and neuronal activity (**left panel**). In ALS, microglia undergo diverse functional changes, which can be protective by clearing aggregates and producing anti-inflammatory and neurotrophic factors, or toxic by producing superoxide radicals, pro-inflammatory cytokines, and factors that transform astrocytes into a neurotoxic reactive state and leading to excessive synaptic pruning (**right panel**).

**Figure 2 biology-12-01307-f002:**
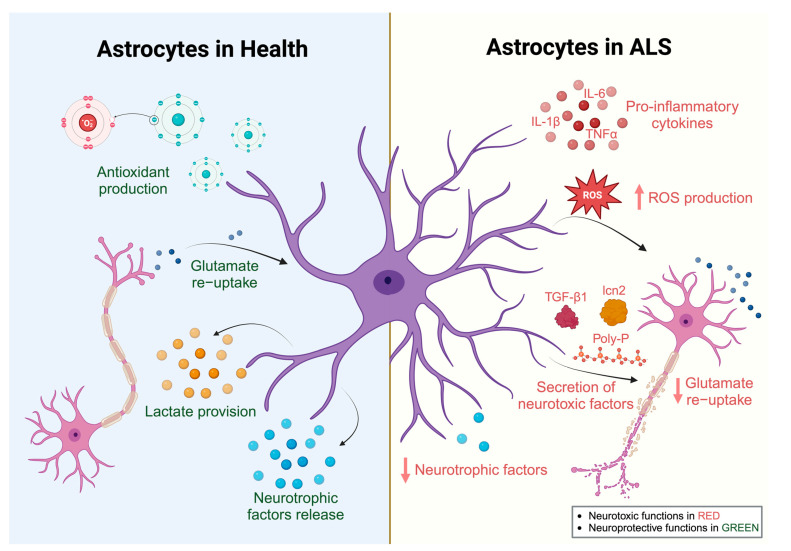
Physiological roles of astrocytes in health and ALS. The physiological functions (**left panel**) of astrocytes under normal healthy conditions which support neuronal health, and these functions include re-uptake of released glutamate from neurons and production of antioxidants, lactate, and neurotrophic factors. The pathological transformation of astrocytes in ALS imposes abnormal functions by producing pro-inflammatory cytokines, superoxide radicals, and neurotoxic factors, decreased glutamate re-uptake, and reduced production of neurotrophic factors, which subsequently lead to neuronal degeneration (**right panel**).

## Data Availability

Not applicable.

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
