# Peer review of "Microglia and Astrocytes in Amyotrophic Lateral Sclerosis: Disease-Associated States, Pathological Roles, and Therapeutic Potential"

_biology, 2023, doi:10.3390/biology12101307_

Round 1

Reviewer 1 Report

The review by You and colleagues focuses on recent publications dealing with the role of microglia and astrocytes in ALS. The majority of the literature cited relates to genetic models of ALS. These account for only 20% of cases.  The remaining cases are likely to be caused by a mixture of genetic and environmental factors. To significantly increase the impact, the review should also consider the evidence for neuroinflammation in relation to environmental stimuli.

Author Response

Comment: The review by You and colleagues focuses on recent publications dealing with the role of microglia and astrocytes in ALS. The majority of the literature cited relates to genetic models of ALS. These account for only 20% of cases.  The remaining cases are likely to be caused by a mixture of genetic and environmental factors. To significantly increase the impact, the review should also consider the evidence for neuroinflammation in relation to environmental stimuli.

Our response: We appreciate your comments and suggestions. Historically, although 90% cases are sporadic ALS, where the patient has no family history of ALS, 10% cases are familial ALS, genetically inherited. Current genetic studies indicates that some of the sporadic cases could be explained by genetic mutations. For example, C9orf72 repeat expansions can be found in 5% of sporadic ALS cases. Still yet, it is undetermined what causes the majority of sporadic ALS cases and what the exact environmental stimuli causes ALS. Therefore, this manuscript focused on reviewing the studies that investigated the pathological roles of microglia and astrocytes from genetic models of ALS. In attempts to add information regarding the role of these cells in sporadic ALS, we have added a new section on human iPSC-derived microglia and astrocytes which includes both familial and sporadic ALS patients.

Reviewer 2 Report

In this review, authors introduced the physiological and pathological role of microglia and astrocyte in neurodegenerative disease, especially in ALS. They also covered the current clinical trials targeting microglia and astrocyte in the ALS treatment. This review is well organized and intriguing, few limitations need to be further addressed. Please see individual comments below.

1.       Does microglia and astrocyte affect other inflammatory cell infiltration such as T cell?

2.       Figure 2 figure legend is not completed.

Author Response

Comment 1: Does microglia and astrocyte affect other inflammatory cell infiltration such as T cell?

Our response: We thank the reviewer for the comments and suggestions. We have added a paragraph explaining the crosstalk between astrocytes/microglia and other peripheral immune cells such as T cells and natural killer (NK) cells.

Comment 2: Figure 2 figure legend is not completed.

Our response: A brief description has been added to both Fig. 1 and Fig. 2 legends.

Reviewer 3 Report

This review provides a useful summary of the field and includes helpful summary images, descriptions of the up-to-date nomenclature recommendations and a good overview of current trials using glial-targeting therapeutics.  I believe it will be a valuable addition to the field after the following additions:

-'reactive microglia' is part of the old terminology that was earmarked for retirement in reference 16.  In lines 197-217, it should be emphasised that these studies are using the old terminology.

-There are a lot of studies that explore the role of astrocytes/microglia in ALS using iPSC-derived human astrocytes and microglia. This is briefly mentioned in line 424 but deserves its own section due to the breadth of studies, increased validity of these models and their ability to examine cell autonomous contributions to ALS.

-reference 35 should be discussed in the SOD1 section given it has used the more modern omics approach to exploring microglia in this model

-to what extent do the currently-approved ALS treatments impact microglial and astrocyte pathology?

-do the authors think that enough is known about what the 'bad' microglia/astrocytes are that these cells can be selectively targetted with a therapeutic without impacting the 'good' ones?

-The first paragraph of the conclusion presented new information (ie the downsides of the new omics-defined microglial states). This would fit better as an addition to the discussion of the new omics approaches in Section 2.1.

-lines 563-565: references required to back up the statement that sex, gender etc impact microglial and astrocytic function.

Author Response

Comment 1: 'reactive microglia' is part of the old terminology that was earmarked for retirement in reference 16.  In lines 197-217, it should be emphasised that these studies are using the old terminology.

Our response: We thank the reviewer for pointing this out. We took great care to try to conform to these suggestions outlined by reference 16 and have made further minor changes to the text regarding this. Additionally, we have added a sentence in the SOD1 section to clarify that many studies have used outdated terms for referring to microglial states and that we will not be using these terms throughout the text.  

Comment 2: There are a lot of studies that explore the role of astrocytes/microglia in ALS using iPSC-derived human astrocytes and microglia. This is briefly mentioned in line 424 but deserves its own section due to the breadth of studies, increased validity of these models and their ability to examine cell autonomous contributions to ALS.

Our response: We thank the reviewer for this suggestion. We have now added a new section regarding the insights gained from human iPSC-derived microglia and astrocytes.

Comment 3: reference 35 should be discussed in the SOD1 section given it has used the more modern omics approach to exploring microglia in this model

Our response: Thank you for pointing this out. We have now included this reference in the SOD1 section.

Comment 4: to what extent do the currently-approved ALS treatments impact microglial and astrocyte pathology?

Our response: Unfortunately, not much is known on this topic given that microglial and astrocytic responses are generally not included as primary or secondary outcome measures, but rather focused on behavioral or lifespan improvement. Furthermore, minimally invasive methods of detecting inflammatory responses in the CNS are limited primarily to PET tracers and there are no good biomarkers for measuring specific microglial or astrocytic states in live patients.

Comment 5: do the authors think that enough is known about what the 'bad' microglia/astrocytes are that these cells can be selectively targeted with a therapeutic without impacting the 'good' ones?

Our response: This is a great question and there is some evidence that specific states can be selectively targeted in preclinical models of ALS (all of which have been referred to in the manuscript), although its translation in patients have yet to be demonstrated. There are two good examples of this targeting ‘good’ microglia, the “disease-associated microglia (DAM)” in which TREM2 was found highly expressed. TREM2 KO have been shown to prevent homeostatic microglia from transitioning to disease-associated microglia in various neurodegenerative diseases including ALS. TREM2 KO in a hTDP-43 overexpression mice results in the depletion of disease-associated microglia, resulting in worsened ALS symptoms. Another example is the use of RIPK1 inhibitors that have been shown to deplete a microglial state termed RIPK1-regulated inflammatory microglia (RRIM) in ALS mice that show upregulation of proinflammatory pathways. In astrocytes, one of the best examples is the depletion of neurotoxic reactive astrocytes in SOD1 G93A mice by knocking out Il1a, Tnf, and C1qa, resulting in a lifespan increase of over 50%. Currently, RIPK1 inhibitors are in phase 2 clinical trials. However, more research is needed in preclinical models to examine whether TREM2 agonists can boost DAM activity in ALS (which has been shown in other neurodegenerative diseases like Alzheimer’s disease), or identify druggable targets that prevent the transition of astrocytes to neurotoxic reactive astrocytes. We believe that more research is needed on this topic and a combination of pharmaceutical/genetic interventions with single-cell omics technologies in preclinical models will provide insight into how we can selectively target ‘good’ or ‘bad’ microglial/astrocytic states and inform future clinical trials. We have added our discussions on this in the conclusion section.

Comment 6: The first paragraph of the conclusion presented new information (ie the downsides of the new omics-defined microglial states). This would fit better as an addition to the discussion of the new omics approaches in Section 2.1.

Our response: We appreciate your comment. In Section 2.1 where we introduce the topic of “disease-associated microglia”, we wanted to mainly discuss what is known about these disease states and how they contribute to different neurodegenerative diseases. We found that the logic flows better in the conclusion section when discussing the limitations that should be considered when conceptualizing these different disease states.

Comment 7: lines 563-565: references required to back up the statement that sex, gender etc impact microglial and astrocytic function.

Our response: Thank you for pointing this out. We have now added appropriate references.

Round 2

Reviewer 1 Report

The authors have only partially addressed the criticisms of this reviewer. However, the parts of the review that have been implemented compensate for the shortcomings.